# Identification of CD114 Membrane Receptors as a Molecular Target in Medulloblastomas

**DOI:** 10.3390/ijms24065331

**Published:** 2023-03-10

**Authors:** Jander Moreira Monteiro, Jaqueline Isadora Reis Ramos, Ian Teixeira e Sousa, Rayana Longo Bighetti-Trevisan, Jurandir Marcondes Ribas Filho, Gustavo Rassier Isolan

**Affiliations:** 1Department of Neurosurgery, Center for Advanced Neurology and Neurosurgery (CEANNE), Porto Alegre 90560-010, Brazil; 2Postgraduate Program, Mackenzie Evangelical College of Parana, Curitiba 81531-980, Brazil; 3School of Dentistry of Ribeirao Preto, University of Sao Paulo, Ribeirão Preto 14040-904, Brazil; 4Pediatric Intensive Care Unit, Conceição Children’s Hospital, Porto Alegre 90560-010, Brazil

**Keywords:** neurosurgery, surgical oncology, molecular targeted therapy, medulloblastoma

## Abstract

Medulloblastomas are the most common solid tumors in children, accounting for 8–30% of pediatric brain cancers. It is a high-grade tumor with aggressive behavior and a typically b poor prognosis. Its treatment includes surgery, chemotherapy, and radiotherapy, and presents high morbidity. Significant clinical, genetic, and prognostic differences exist between its four molecular subgroups: WNT, SHH, Group 3, and Group 4. Many studies seek to develop new chemotherapeutic agents for medulloblastomas through the identification of genes whose expressions are new molecular targets for drugs, such as membrane receptors associated with cell replication. This study aimed to assess the association of CD114 expression with mortality in patients with medulloblastoma. Databases from the Medulloblastoma Advanced Genomics International Consortium (MAGIC) were analyzed, focusing on the expression of the CD114 membrane receptor in different molecular types and its possible association with mortality. Our findings showed different CD114 expressions between Group 3 and other molecular groups, as well as between the molecular subtypes SHH γ and Group 3 α and Group 3 β. There was no statistically significant difference between the other groups and subtypes. Regarding mortality, this study did not find statistical significance in the association between low and high CD114 expressions and mortality. Medulloblastoma is a heterogeneous disease with many subtype variations of its genetic and intracellular signaling pathways. Similarly to this study, which could not demonstrate different CD114 membrane receptor expression patterns between groups, others who sought to associate CD114 expression with mortality in other types of cancer failed to establish a direct association. Since many indications point to the relation of this gene with cancer stem cells (CSCs), it may be part of a more extensive cellular signaling pathway with an eventual association with tumor recurrence. This study found no direct relationship between CD114 expression and mortality in patients with medulloblastoma. Further studies are needed on the intracellular signaling pathways associated with this receptor and its gene (the CSF3R).

## 1. Introduction

Medulloblastomas are the most common solid tumors in children, accounting for 8–30% of pediatric brain cancers [1,2]. According to the World Health Organization’s (WHO) classification, it is a type of high-grade tumor, which implies aggressive behavior and typically poor prognosis. Its treatment also presents high morbidity. 

Previously known as cerebral spongioblastoma, Harvey Cushing and Percival Bailey first named it medulloblastoma in 1925 [3]. The former denomination was abolished to dispel the idea of a glial origin. Medulloblastomas are tumors that are derived from the primitive neural tube that commonly arise in the posterior fossa and tend to send metastasis foci via cerebrospinal fluid dissemination, which may cause hydrocephalus.

Medulloblastomas occur predominantly in the first decade of life. However, there is no definition in the medical literature about the peak incidence, which varies between 3 and 9 years old, just as there is no well-established predisposing factor. Nonetheless, it may be associated with rare familial syndromes such as Turcot, Gorlin, and Li-Fraumeni [3].

The treatment of medulloblastomas involves surgery, radiotherapy, and chemotherapy. Despite significant advances and increased survival after diagnosis, such a combined invasive approach may be associated with higher morbidity and severe neurological sequelae. Regarding prognosis, patients can be grouped into “low risk” and “high risk” depending on three factors: age, metastatic spread at the time of diagnosis, and incomplete surgical resection. Despite good prognosis in so-called “low-risk” tumors, other factors can be associated with these patients’ prognoses, such as molecular analysis [2].

The research for improvements in medulloblastoma treatment aims for new chemotherapeutics that are directed to specific molecular targets, such as membrane receptors associated with cell replication. The search for genes that are related to medulloblastomas can identify prognostic markers, which can help clarify why patients with similar stratifications occasionally evolve in different ways.

One of the molecular targets under study is the granulocyte colony-stimulating factor receptor (G-CSF-R), also known as CD114 (cluster of differentiation 114), encoded by the CSF3R gene. Its expression has already been implicated in the pathogenesis of several tumor types, including ovarian carcinomas, bladder cancer, and skin cancer [4,5,6]. CD114+ cancer cells have displayed the ability to self-renew, generate differentiated progeny, and recapitulate a subpopulation of heterogeneous tumor cells. In medulloblastoma cells, it has been observed that a subpopulation of CD114+ cells presented altered growth, chemoresistance, and responsiveness to the granulocyte colony-stimulating factor (G-CSF): a drug sometimes used in cancer treatments, including medulloblastomas [7]. 

If its role in the pathogenesis of medulloblastomas is confirmed, CD114 could be used as a therapeutic target to develop new drugs or even as a diagnosis and prognosis marker of the disease.

### Objectives

Primary: Evaluate the association of CD114 expression with mortality in patients with medulloblastoma.

Secondary: Assess the difference in CD114 expression among molecular subtypes of medulloblastomas.

## 2. Results

In this study, we included data from 763 patients. Samples of primary tumors from patients diagnosed with medulloblastoma were analyzed. Table 1 presents the descriptive data regarding the age, gender, and survival of these patients that was distributed for each molecular tumor type.

Classically, medulloblastomas have been classified into histological subtypes. However, this classification has been losing relevance since the change proposed by the World Health Organization (WHO) in the 2016 Classification of Central Nervous System (CNS) Tumors. Recently, molecular classification has been used for these tumors, and several studies have deepened our knowledge about the genomics of this disease. Table 2 shows the relationship between the histological (rows) and molecular (columns) classification of medulloblastomas. This table demonstrates how a single histological subtype can present a broad spectrum of biological behavior, from benign to very aggressive. The classical subtype, for instance, was represented by a slight majority in the Group 4 molecular subtype. However, the remainder was divided relatively evenly among the three other molecular subtypes. Likewise, the histological large cell/anaplastic (LCA) subtype showed a very similar molecular classification among the three different groups: SHH, Group 3, and Group 4.

To define whether the CSF3R gene expression followed a normal distribution, we performed the Kolmogorov–Smirnov test, which confirmed the null hypothesis that it did not. This analysis can be seen in Figure 1 in a Q–Q plot.

After defining that the gene expression did not follow a normal distribution, we performed the Kruskal–Wallis test to determine whether the different patterns of the CSF3R gene expression among the groups were statistically significant, which was confirmed (*p* < 0.001). Subsequently, we applied a post hoc Dunn’s test to assess between which groups this difference was relevant. The test showed that the tumors in Group 3 had a CSF3R gene expression pattern which differed from the other groups (WNT, SHH, and Group 4). Figure 2 shows the relationship between the patterns of the CSF3R gene expression among the four main molecular types of medulloblastomas, highlighting the difference between the tumors in Group 3 and the others.

An analysis of CSF3R gene expression among the molecular groups of medulloblastomas was once again performed (Kruskal–Wallis test, followed by Dunn’s post hoc test), but this time, the molecular subgroups proposed by Cavalli et al. were used [8]. The results show that the SHH gamma subgroup presented a different expression pattern compared to Group 3 alpha, Group 3 Beta, and SHH alpha subgroups (*p* < 0.001). Figure 3 shows the relationship of CSF3R gene expression patterns among the 12 molecular subgroups of medulloblastomas.

In addition, we analyzed the impact of CSF3R gene expression on patient survival. The expression was characterized as “low” and “high” using the median value as a criterion, and the time interval was measured in years. Figure 4 shows the results in Kaplan–Meier curves, displaying each subgroup separately alongside the total sample. None of the analyses reached statistical significance (*p* < 0.05).

Similarly, we analyzed the impact of CSF3R gene expression on mortality using different values as the parameters for “low” and “high” expressions. In Figure 5, the 25th percentile was used as a criterion. The values below it were considered low gene expressions, while those above were high expressions. On the other hand, Figure 6 shows the analysis that considered the 75th percentile as the cut-off point. The time interval was also measured in years, and the results are expressed in Kaplan–Meier curves. Both analyses also did not reach statistical significance (*p* < 0.05).

## 3. Discussion

Medulloblastomas are a heterogeneous disease. Although their various subtypes are generally identified as a single disease, it is currently known that each subgroup presents oncogenesis, genetic alterations, and distinct biological behavior [8,9]. Nowadays, genomic analyses are the gold standard for categorizing tumor subtypes, and, despite still showing some overlap between groups, this happens less than in classical histology. Since the 2016 WHO CNS tumors classification, medulloblastomas have been divided into four subgroups: WNT, SHH, Group 3, and Group 4 [10]. However, precisely how heterogeneous these groups may be has yet to be discovered. Subclassifications have been proposed for each of the four molecular types, with remarkably distinct clinical or cellular characteristics [11,12]. We could see the genetic variation within the types and subtypes of these tumors when the CSF3R gene expression was analyzed. Although Group 3 showed a different expression pattern from the other molecular types, there was no statistical significance when the other groups were compared with one another. Likewise, among the subgroups proposed by Taylor et al. [12], SHH γ showed a different expression pattern from Group 3 α and Group 3 β with statistical significance. These data are compatible with the literature and reinforce the heterogeneity of the disease and its molecular types, indicating how far the understanding of this disease still has to progress.

The granulocyte colony-stimulating factor (G-CSF) was first identified in bone marrow cells and derivatives. It is a glycoprotein that stimulates, through its ligand on the cell membrane (CD114), the growth, differentiation, and activation of defense cells (chemotaxis, degranulation, and phagocytosis) [5]. It is produced mainly by monocytes and macrophages but also by fibroblasts, endothelial cells, and bone marrow stroma. Its production is significantly increased in the presence of inflammatory activity [13,14]. However, cancer cells of various origins exhibit a stimulatory response to the presence of G-CSF, such as malignant tumors of the bladder, liver, oral cavity, skin, neuroblastomas, and leukemia, as has been shown [4,5,14].

Cancer cells, such as defense cells, have an autocrine loop of G-CSF production. The transfection (i.e., the introduction of genetic material into a cell) of the CSF3R gene into human osteosarcoma cell lines makes these cells self-stimulatory and able to survive and proliferate independently of the extracellular matrix, which is a significant evolutionary gain for carcinogenesis and cancer perpetuation [5]. Although this pathway is not yet as well established as in hematopoietic cells, both the growth, proliferation, and differentiation response, as well as the production of the stimulating factor, have already been proven for some tumor types [4]. Chakraborty and Guha [4] demonstrated that, in bladder cancer, the increased expression of the G-CSF receptor is associated with the overexpression of the surface protein Integrin beta-1, which has the function of cell adhesion. In this type of cancer, such an association is related to the greater local invasiveness of the tumor.

Moreover, the overexpression of CD114 is associated with a higher rate of tumor growth and migration, in addition to decreased apoptosis [4]. Kumar et al. [15] found similar results studying CD114 expression and ovarian cancer. Besides its expression being associated with greater disease aggressiveness, the exogenous use of G-CSF resulted in more extensive migration and a lower apoptosis/tumor cell death rate, in addition to the activation of tumor perpetuation pathways [15]. Understanding CSF3R gene expression is, therefore, crucial to understanding each tumor type’s oncogenesis and, eventually, seeking ways to block it.

Hirai et al. demonstrated that in squamous carcinomas of the oral cavity, actinic keratosis, and Bowen’s disease (a variant of squamous cell carcinoma), the well-known high expression of CSF3R was not related to increased mortality [5]. In their study, Hirai et al. compared CSF3R gene expression with the Ki-67 nuclear antigen marker: a relevant prognostic factor in oncology. They showed that CSF3R gene expression, either increased or decreased, had no impact on the Ki-67 value for any of the three diseases [5]. The findings are compatible with the results of this study regarding medulloblastomas.

Hirai et al. [5] also hypothesized an association with local cancerous invasiveness in skin tumors, similar to bladder cancer. Elevated CD114 activation is associated with increased activity of the matrix metalloproteinases (MMPs), which are responsible for the degradation of the basal lamina of the skin; this could facilitate local tumor dissemination. Other genes are involved in MMP production and activity, such as the one that encodes, in the cell membrane, the epidermal growth factor receptor (EGFR), which is named by the same acronym (EGFR). The activation of the EGFR gene has also been associated with skin basement membrane degradation through MMP activity. However, the relationship between these two genes, CSF3R and EGFR, is not yet well established. As the mechanisms of action of the CSF3R gene in cancer cells are still not well defined, it may be only part of a pathway in which multiple mechanisms interact with an unfavorable prognosis as the final result. Even so, the CSF3R gene, alone, could not provide a direct association [5].

In this study, the Kaplan–Meier curve for survival analysis showed no statistical difference between the groups with high or low CSF3R expression. These data are consistent with the few studies on this gene regarding survival in cancer patients. However, this is the only study to date in which this analysis has been performed for patients with medulloblastoma. In an attempt to establish reference values for the division of the groups to be analyzed (one with high and the other with low values), we applied the median, 25%, and 75% percentiles. None of the analyses showed statistical significance regarding mortality. It may be due to a sampling error since some analyses, especially of subgroups, had a p-value close to 0.05, established in this study as the required value to rule out the null hypothesis (that is, that the results found were due to chance). Another difficulty in data analysis was measuring the gene expression, which was calculated from primary tumor samples. Their numerical expressions presented considerable variations, and the graphic representation was unfeasible. Therefore, we transformed these measurements into binary logarithms, which allowed for analysis. Thus, it was possible to affirm that the different groups had different CSF3R gene expression patterns, but it could not be said that one group had a necessarily higher expression than another. The ideal approach for such gene expression measurement would be through the cell lines of tumor cells in a controlled environment, in which it would be possible to define what would be increased and to establish the cut-off point for the analysis of the groups: a reference that the database does not allow.

A crucial aspect of identifying CSF3R gene activity in medulloblastomas is understanding the effects of oncologic treatment on the disease. Patients undergoing chemotherapy and radiotherapy, two of the mainstays of medulloblastoma treatment, are at high risk of developing bone marrow aplasia. Among children, who are not always eligible for radiotherapy treatment, protocols sometimes use higher doses of chemotherapy [16]. In such cases, in situations of severe neutropenia and its complications, the use of G-CSF is indicated to stimulate the production of defense cells [9,14,17]. However, studies on skin and bladder malignancies have demonstrated, in vitro, the increased replication, differentiation, and migration of tumor cell lines after stimulation with exogenous G-CSF. In addition, blocking CD114 through neutralizing antibodies has an inhibitory effect on the proliferation and migration of these cancer cells [4,5,13]. Staar et al. [18] described how the loco-regional control of head and neck tumors in patients who received prophylactic treatment with G-CSF was worse than in those who did not. Russell and Shohet [19] concluded that there was insufficient evidence for the use of high doses of G-CSF to promote bone marrow recovery after myeloablative chemotherapy. Still, the use of G-CSF is customary in oncology. All these findings suggest that, even if there is not a direct effect of the CSF3R gene on the mortality of patients with medulloblastoma, other factors, such as treatment, may influence this outcome [4,5,13]. There is already enough evidence on the stimulatory effect of G-CSF in several types of cancers for its use in chemotherapy routines to be re-evaluated [14].

Cancer stem cells (CSCs) are multipotent neoplastic cells with the ability to self-renew, differentiate into various tumor cell types, and eventually reconstitute an entire tumor cell population. This cell type has already been identified in some types of cancer, including medulloblastomas [20,21]. When present, the tumors tend to have a worse prognosis, stronger resistance to chemotherapy treatment, and a higher rate of relapses [22,23]. Several cell surface markers for CSCs have already been proposed in the literature, including CD133, ALDH, CD44, EpCAM, and CD271 [14]. However, due to the vast heterogeneity of these cells within an already quite heterogeneous group of cancer cells, their exact identification and the mechanisms through which these cells’ signaling pathways function are still unclear. There is even the possibility that the same marker acts differently in distinct types of cancer [9,21].

Paul et al. described CD114 as a possible CSC marker in medulloblastomas in 2020 [9]. Studies have already described the CD114 receptor as a CSC marker in neuroblastomas, Ewing’s sarcomas, and melanomas, demonstrating a connection to recurrence [14]. In medulloblastomas, CD114+ cells exhibit all the characteristics expected in CSC: their presence in a small and undifferentiated number of cells; stronger resistance and higher percentage growth after chemotherapy treatment; and the potential for differentiation such as pluripotent embryogenic cells, in addition to the responsiveness of G-CSF granted by the receptor. In these cells, the gene and signaling pathways that are common to several medulloblastoma lineages have been identified, such as NRP1 [24], MSI1 [25], TWIST1 [26], MYCN [27], and OX2 [28].

New therapies targeting CSCs may positively impact cancer treatment, including medulloblastomas. CD114 is a molecular target that is under study, but there are still no drugs in trial for CSCs. The study of CSCs presents substantial difficulties and limitations. The CSC population in tumor samples is small, and the culture media and conditions of the in vitro analysis of these cells impact their phenotypes, making it challenging to analyze and extrapolate laboratory results to patients [29]. Regarding medulloblastomas, the heterogeneity of their different molecular groups and subgroups and their various cells of origin are an additional complication factor in understanding CSCs and proposing new treatments. Diverse gene expressions in each subgroup make it necessary to know each molecular subgroup extensively to search for a possible common signaling pathway, which may not even exist [9]. Only through complete knowledge of the intracellular CD114 pathway will we be able to understand its actual impact on mortality and relapses and then propose treatments that are personalized to its molecular signature. Understanding CSC and relapse in medulloblastomas represents the next revolution in treating the disease.

## 4. Materials and Methods

This study was carried out through the analysis of a public database that was elaborated by the worldwide consortium for the study of medulloblastomas, known as the Medulloblastoma Advanced Genomics International Consortium (MAGIC). The consortium has samples from more than 2000 medulloblastomas from centers of excellence in treating brain tumors, mainly from North America (Canada and the United States of America) but also from Europe, Asia, and South America (including Brazil). The MAGIC database is available for download at: https://ncbi.nlm.nih.gov/geo/query/acc.cgi?acc=GSE85217 (accessed on 15 June 2019). 

### 4.1. Sample Selection and Data Collection

The database that was used in this study was created using information from patients whose tumor samples were analyzed for CSF3R gene expression. In total, 763 samples were included in the database. We selected data on the study population (epidemiological data) and gene expression. The analyzed data related to gender, age, survival and mortality, histological and molecular classifications of the tumors, and the expression of the CSF3R gene.

### 4.2. Measurement of Gene Expression

To measure CSF3R gene expression, an in silico analysis of products transcribed by the tumor cell (messenger RNAs, ribosomal transporters, and microRNAs), also known as transcriptome, was performed.

### 4.3. Statistical Analysis

The response variable evaluated in this study was the CSF3R gene expression. Due to discrepant values resulting from the gene expression in the database, the variable was transformed into a binary logarithm (log2) to make the data homoscedastic (of constant variation). To verify the normality of data distribution, we applied the Kolmogorov-Smirnov test and the graphical visualization through a Q-Q plot. As it did not meet the assumptions of normality, the effect of molecular subgroups of medulloblastoma on gene expression was verified using the Kruskal–Wallis non-parametric test, followed by Dunn’s post hoc test to compare the distributions among each molecular subgroup.

In addition, we performed a Kaplan–Meier (KM) analysis to estimate median survival times and their respective confidence intervals. The KM method is a non-parametric statistic that is used to estimate the probability of survival from observed survival times. The survival curves on the KM plot provide an association between time (in years) and the probability of survival, which can be used to estimate a patient’s median survival time. We also performed the Log-rank test to determine whether a factor assessed in the medical record could be statistically associated with differences in the survival curves. This non-parametric test assumed a null hypothesis that there was no difference in survival between the two tested groups, with no prior assumptions about survival distributions. Essentially, the Log-rank test compares the observed number of events in each group to what would be expected if the null hypothesis were true (that is if the survival curves were identical): a statistic similar to the chi-square test.

## 5. Conclusions

This study found no direct association between CD114 membrane receptor expression and mortality in patients with medulloblastoma. However, further studies on the intracellular signaling pathways associated with this receptor and its gene (CSF3R) are needed. Additionally, future research is required to investigate its interaction with other genes that are associated with the pathogenesis of these tumors. Current knowledge about the CSF3R gene is limited, and many other questions remain unanswered regarding its role in tumor recurrence and patient survival.

Although we could find statistically significant differences between some molecular groups and subgroups of medulloblastoma in most analyses, the difference was not relevant. Therefore, it remains difficult to characterize these tumors’ gene expression patterns. Further studies on the genomics of medulloblastomas need to be undertaken to understand them better and individualize their treatment.

## Figures and Tables

**Figure 1 ijms-24-05331-f001:**
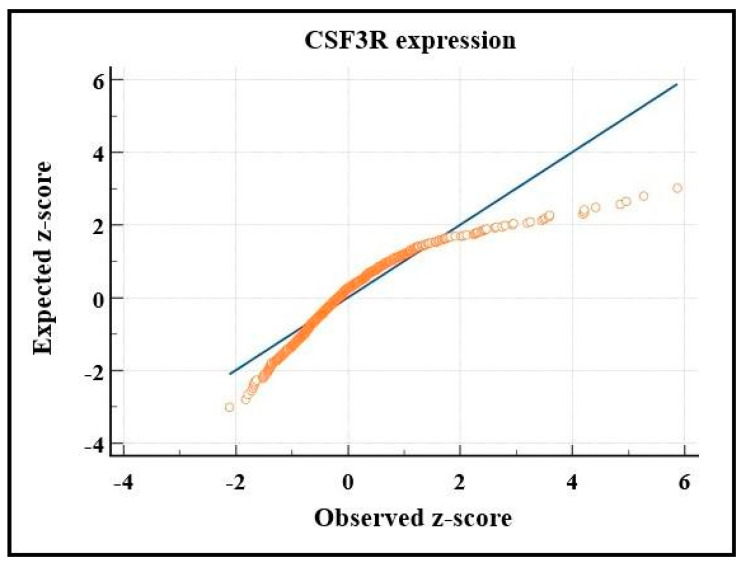
CSF3R expression distribution pattern. The straight line represents the reference for normal distribution, while the orange marks represent the gene expression data. Source: the authors. Compiled using MedCalc^®^ version 20.115 (MedCalc Software Ltd., Ostend, Belgium).

**Figure 2 ijms-24-05331-f002:**
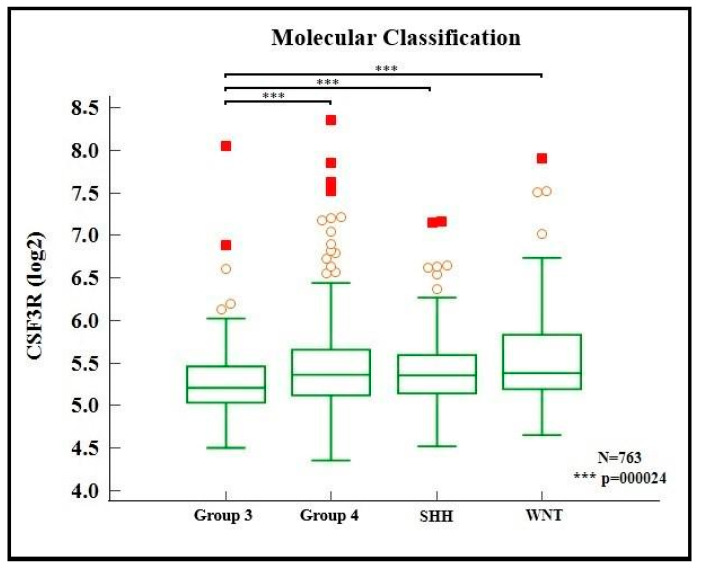
CSF3R expression in medulloblastoma molecular groups. Red circles and squares represent, respectively, outside and far out values (larger than the upper quartile plus 1.5 and 3 times the interquartile range, respectively). Source: the authors. Compiled using MedCalc^®^ version 20.115 (MedCalc Software Ltd., Ostend, Belgium).

**Figure 3 ijms-24-05331-f003:**
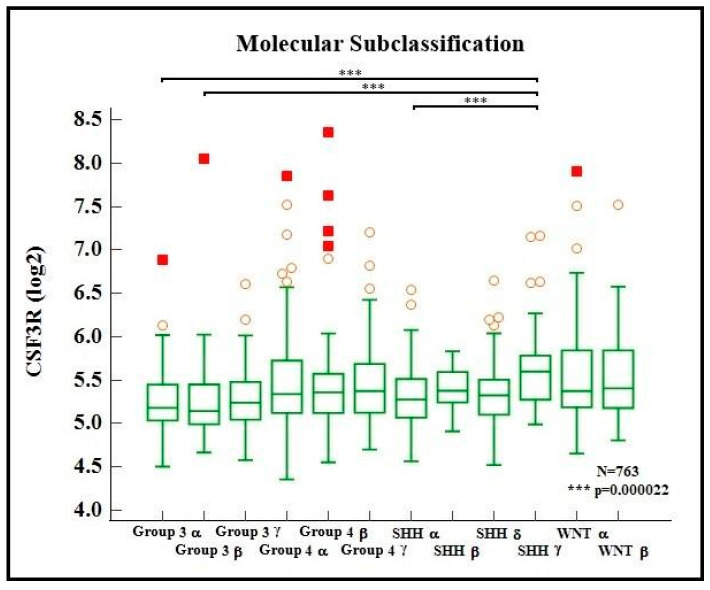
CSF3R expression in the molecular subgroups of medulloblastomas. Red circles and squares represent, respectively, outside and far out values (larger than the upper quartile plus 1.5 and 3 times the interquartile range, respectively). Source: the authors. Compiled using MedCalc^®^ version 20.115 (MedCalc Software Ltd., Ostend, Belgium).

**Figure 4 ijms-24-05331-f004:**
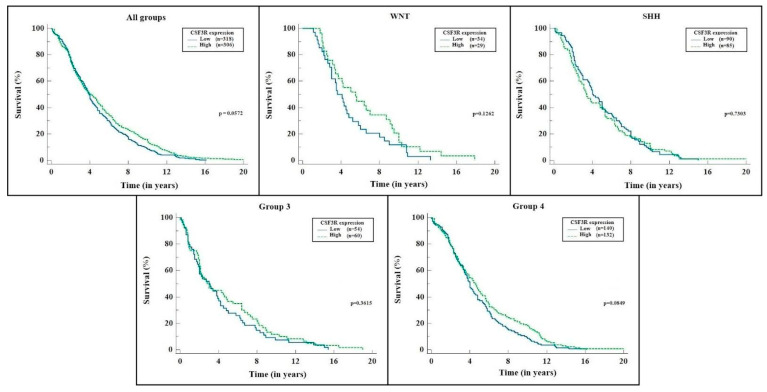
CSF3R expression × survival (median) for each molecular group and total sample. Source: the authors. Compiled using MedCalc^®^ version 20.115 (MedCalc Software Ltd., Ostend, Belgium).

**Figure 5 ijms-24-05331-f005:**
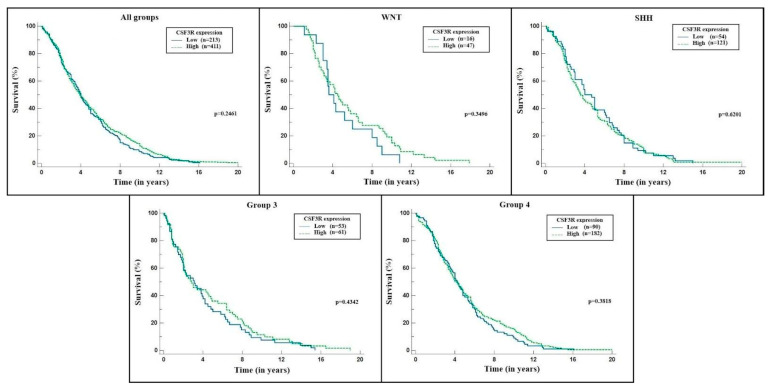
CSF3R expression × survival (25th percentile) for each molecular group and total sample. Source: the authors. Compiled using MedCalc^®^ version 20.115 (MedCalc Software Ltd., Ostend, Belgium).

**Figure 6 ijms-24-05331-f006:**
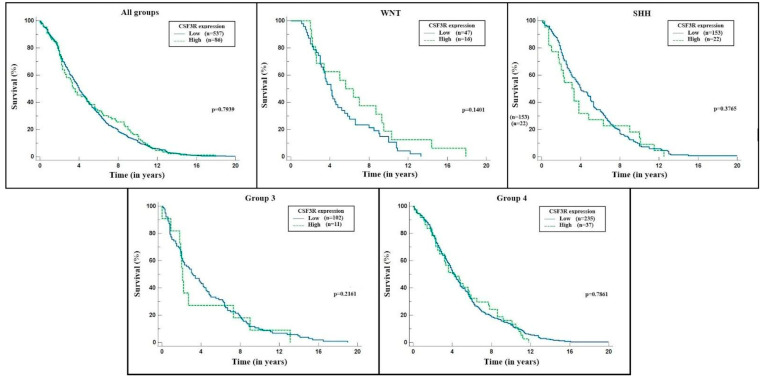
CSF3R expression × survival (75th percentile) for each molecular group and total sample. Source: the authors. Compiled using MedCalc^®^ version 20.115 (MedCalc Software Ltd., Ostend, Belgium).

**Table 1 ijms-24-05331-t001:** Study population.

Variable		WNT	SHH	Group 3	Group 4	Total Sample
Age (years)		10.8 (8.0–16.0)	8.8 (2.1–21.5)	5.1 (3.1–8.0)	8.0 (5.8–11.0)	8.0 (4.8–12.0)
Gender	Male	29	128	99	216	472
	Female	35	82	38	92	247
Gender ratio		0.8	1.6	2.6	2.3	1.9
Survival (years)		4.2 (2.7–8.3)	3.9 (2.1–7.0)	3.0 (1.4–6.7)	4.2 (2.3–6.8)	4.0 (2.1–6.9)

**Table 2 ijms-24-05331-t002:** Histological X molecular diagnosis of medulloblastomas.

Variable	WNT	SHH	Group 3	Group 4	Total Sample
Classical	40 (10.3%)	78 (20.2%)	68 (17.6%)	201 (51.9%)	387 (100%)
Desmoplastic	5 (4.6%)	73 (67%)	8 (7.3%)	23 (21.1%)	109 (100%)
MBEN	0 (0%)	10 (55.6%)	2 (11.1%)	6 (33.3%)	18 (100%)
LCA	5 (7%)	20 (28%)	25 (35%)	22 (30%)	72 (100%)

## Data Availability

Not applicable.

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
