# Peer review of "Identification of CD114 Membrane Receptors as a Molecular Target in Medulloblastomas"

_ijms, 2023, doi:10.3390/ijms24065331_

Round 1

Reviewer 1 Report

In this article, Monteiro et al. tried to identify the significance of CD114 in medulloblastoma. I have some suggestions for this manuscript.

1. It needs to be clarified why CD114 was selected for this study. The authors should have provided a rationale justifying CD114.

2. Authors should have cited some landmark studies such as (PMID: 28545823)

Author Response

Dear Reviewer,

thank you for taking your time to review our paper.

sorry it took me so long to answer you.

Addressing your questions, CD114 was chosen based on its possible relation with recidive mechanism already in study in other cancer types. Also, it may be associated with cancer stem-cells.  Although it was not the main focus on the introduction, we tried to follow a line of thinking, making the text cohesive and pleasurable to read and reaching the idea at its end. It is thoroughly discussed in the discussion section. 

We agree that the article PMID: 28545823 is outstanding. The point is that the classification is not the focus of our work and, as molecular subclassification is a new developing area, we felt that addressing more than one subclassification would be a shift from the main goal.

Thank you so much for you time and considerations

Jander

Reviewer 2 Report

In this study the authors reported that there is no direct association between CD114 membrane receptor expression and mortality in patients with medulloblastoma after a retrospective analysis from a database. 

They concluded that further studies on the intracellular signaling pathways associated with this receptor and its gene are needed.

While the methodology of this manuscript is valuable, there is no real novelty in this research.

As authors said "The current knowledge about gene of this receptor is limited, and many other questions remain unanswered". Please provide a limitation study paragraph and an English language check.

Author Response

Dear reviewer, thank you for your time and attention on our paper

sorry it took me so long to answer you.

Addressing your considerations: although our results were not positive, there were only 2 other published articles in PubMed about CD114 and medulloblastomas, none of the looking for a direct relation between CD114 and mortality. We believe that, even though negative, our study is actually new information and could guide following researches in this matter in new paths other than this direct one.

We added a limitation paragraphs to the end of discussion.

thank you for your time and considerations

Jander

Round 2

Reviewer 1 Report

The authors of the study have justified their choice of CD114 by citing its potential link to relapse in other types of cancer and its association with cancer stem cells. However, while various molecules have been examined in different cancer types and stem cells, it is not apparent why CD114 was specifically selected for this investigation. Furthermore, since CD114 is also expressed in macrophages and microglia during inflammation and tumor-associated macrophage/microglia are found at a higher percentage in several brain tumors. It is unclear whether the expression observed in the study is exclusive to the cancer cells/ cancer stem cells.

The authors have utilized a publicly accessible dataset derived from tumor tissue, which encompasses a heterogeneous cell population. As a result, to reinforce their results, it would be appropriate for the authors to employ more cancer stem cell-specific markers since numerous cell types in the tumor microenvironment share CD114.

Author Response

Dear reviewer

thank you for your comments.

As stated in the manuscript, the study aimed to correlate the CD114 cell marker with medulloblastomas. This study is part of a wider study based on a variety of cell markers in brain tumors. This study was done based on a published brain tumor database.  As said in the article, brain tumor samples were collected in many reference centers around the world and sent to transcriptome analyses. Although the method of sample analysis may be imperfect (other cell types present in the sample may also express the cell marker we aimed), we believe it is valid method of analysis. Tumor samples are used in day-to-day clinical practice in immunohistochemical pathological diagnosis. At least, the data collection method can correlate to practical medicine and, therefore, are a "good enough" method of information extraction. There may be more advanced ways to extract transcriptome information and, possibly, generate the "most accurate" results. However, it outsources and deviates from the main purpose of our study.

The CD114 is associated with other cancer types and with cancer steam-cells. Although our results were negative, we believe that this negative study can help future research in methodological changes and/or new objectives while studying this cell marker. There is, for sure, a lot more research to be done in this field. However, we tried to analyze a simple correlation (CD114 - Medulloblastoma) with our limited data. Presenting other markers or associating with cancer steam-cells would deviate from our main focus. We went for a simple and objective study. Although we did not achieve the positive results initial analyses made we hope for, this study helps to fill a void in current literature about this specific topic.

Jander Monteiro